# Medicine information helpline after hospitalization–a randomized trial: Impact on patient satisfaction, patient concerns about medicines and clinical outcome on patient safety

**Karianne Wilhelmsen Fjære**[1☯], **Tim Emil Vejborg**[1☯], **Lene Colberg**[1☯], **Cecilia Strøjer Ulrich**[1‡], **Lars Pedersen**[2‡], **Ann Kathrin Demény**[3‡], **Joo Hanne Poulsen**[1‡], **Helle Byg Armandi**[1‡], **Marianne Hald Clemmensen**[1☯]*

1 Medicines Information Center, The Hospital Pharmacy, Capital Region of Denmark, Bispebjerg Hospital, Copenhagen NV, Denmark, 2 Department of Respiratory Medicine, Capital Region of Denmark, Bispebjerg Hospital, Copenhagen NV, Denmark, 3 Emergency Department, Capital Region of Denmark, Bispebjerg Hospital, Copenhagen NV, Denmark

☯ These authors contributed equally to this work.
‡ These authors also contributed equally to this work.
* haldlarsen@hotmail.com

## Abstract

### Background and aim

Hospitalization often leads to changes in patients' medicine which challenges a safe medication use after discharge. Medicine information helplines (MIHs) can be valuable for patients in overcoming these challenges. This study evaluates patient satisfaction with a newly established Danish hospital-based MIH for discharged patients. The MIH is operated by experienced pharmacists and a pharmacy technician, and the study explores how the service affects the patient's concerns and perception of safety in relation to their medication, followed by an assessment of the clinical impact of MIH on patient safety.

### Method

A randomized controlled study design was used in the present study. The study was registered at clinicaltrials.gov with the identification number NCT03829995. Participants were randomized 1:4 (50:200) into a control- and intervention group. Participants in the control group were offered standard care and those in the intervention group were offered access to the MIH. A telephone interview performed 2–4 weeks after discharge assessed patient satisfaction with the helpline and patient's feeling of safety in relation to medicine use (primary outcome). Data were analyzed using a Mann-Whitney U test. After case handling of each enquiry to the MIH, the cases were assessed with regard to medication-related problems (MRPs) and clinical impact of the MIH service was assessed (primary outcome).

**Data Availability Statement:** All relevant data are within the paper and its Supporting Information files.

**Funding:** MHCL received one grant supporting the research presented in the article. No other authors received any grants or awards. The grant was received from The Danish Research Unit for Hospital Pharmacy, Amgros I/S, Copenhagen, Denmark. https://amgros.dk/om-amgros/samarbejdspartnere/nationalt-samarbejde/ The funders did not play any role in study design, data collection and analysis, preparation of the manuscript or in the decision to publish the data. The grant only finansed a small part of the total costs of the study.

**Competing interests:** The authors have declared that no competing interests exist.

## Results

A total of 250 participants were included in the study and 152 participated in the telephone interviews (33 control and 119 intervention). Thirty-seven questions were enquired by 26 participants to the MIH. Of these, 8 were requested before the telephone interviews and these patients all expressed a high satisfaction with the MIH (score 4.57 +/- 0.73 on a 5-point scale). Most patients offered access to the MIH expressed that it increased the sense of safety in relation to their medicines (79%). However, comparing the control- and intervention group with regard to patient concerns and feeling of safety in relation to medicine use no differences were found. Evaluation of the enquiries revealed at least one MRP per enquiry, and in most cases the advice given were assessed to have a high- or moderate clinical significance.

## Conclusion

The MIH was appreciated by the participants, indicating that the MIH could be a valuable service for discharged patients in improving the sense of safety in relation to medication and alleviating MRPs. Providing easy access for patients to medicine information may contribute to patient safe medicine use after discharge.

## Introduction

Healthcare systems are becoming more and more complex, and patients must navigate between hospitals and the primary sector. Transition from hospitals to private homes is known to challenge patient safety [1–3]. Recent studies highlight that adverse events related to cross-sectorial care are increasing of which many can be attributed to errors in the medication process [1–3]. During hospitalization patients often experience changes in their medication, which can lead to confusion and insecurity about the new or changed medicine after discharge [1,3]. Further, patients report that they did not receive important information about their medication upon hospital discharge [2,4,5]. It is estimated that up to 65% of patients discharged from a hospital subsequently experience medication-related problems (MRPs) defined as events or circumstances related to a patient's medication that can adversely affect patients' health status [3].

Importantly, it has been shown that patient education and counseling can reduce the risk of medicine related adverse events and rehospitalization, and patients have been identified as a key resource in improving medicines safety in cross-sector situations [1,2,4]. Thus, there is a need to ensure that patients receive the relevant support, information and education related to their medicine.

Medicine Information Helplines (MIH) can play an important role in supporting patients in a safe medication after hospitalization [1,2]. Several studies have shown that patients in general express a high satisfaction with MIH services and the information they have received about their medicine [1,5–8]. In the UK, MIH services were established in the early 1990s enabling patients discharged from the hospital direct access to communication with a pharmacy professional to improve the medication safety and to reduce medication errors [6]. Similar services have been established in countries like Norway, Sweden and Germany [9–11]. MIH could also support a better medication adherence by empowering the patients through provision of knowledge and information about their use of medicines [1,6,7,12,13].

Up until 2018, discharged patients in Denmark had no access to a MIH. Patients may contact the general practitioners, community pharmacies or the respective hospital ward with medicine-related questions [9]. However, this is not always suitable, as the new medicine may be used in hospitals only, the general practitioners may be unavailable, the community pharmacy does not have access to the discharge summary from the hospital or the hospital staff might be busy taking care of admitted patients. Therefore, the Danish hospital-based service, MIH, was established in 2018 for discharged patients in the Capital Region of Denmark [14]. This study is to our knowledge the first to explore if addition of MIH affects patient's concerns and perception of safety about medicines use as compared to standard care with limited medicine information support after discharge. Furthermore, the aim was to evaluate patient satisfaction with the new MIH followed by an assessment of the clinical significance of MIH on patient safety.

## Method

### Setting

The MIH was operated by experienced pharmacists and a pharmacy technician from the Medicine Information Centre (MIC) at the Hospital Pharmacy at Bispebjerg Hospital in the Capital Region of Denmark. Staff was trained in clinical pharmacy and drug information activities. The MIH team consists of 12 medicine information pharmacists and a pharmacy technician, who have been providing medicine information between 5–25 years in the MIC. The helpline is accessible on all weekdays from 8 am to 3.30 pm.

An acute care setting and a non-acute inpatient setting were purposively chosen to obtain possible variations in the clinical impact related to the MIH, and the impact on patient satisfaction, concerns and perception of safety in relation to their medicine after discharge. Thus, the Emergency Department and the Department of Respiratory Medicine at Bispebjerg Hospital were selected. Bispebjerg Hospital is one of the hospitals in the Capital Region of Denmark. The hospital serves as a community hospital for approximately 400,000 citizens. The Emergency Department has around 40 beds while the Department of Respiratory Medicine has 38 beds [15].

### Enrollment of participants

A total of 250 patients were enrolled in the study from July 2018 to May 2019. The patients were enrolled from the two selected hospital wards by the first three authors from the MIC. All admitted patients at the two wards were assessed and eligible participants were subsequently approached for enrollment. Eligible participants were 18 years or older, spoke and understood Danish, and were scheduled for discharge to their private homes within a few days after enrollment. Exclusion criteria included severe dementia diagnosis, terminally illness, isolation precautions, aphasia, suicidal or patients in custody. Length of stay and whether the patient used medicines or not were not exclusion criteria.

Participation in the study was voluntary and all patients were given oral and written information about the study, publication plans and his or her right to withdraw from the study at any time without further consequences. All participants signed a consent form and patients' participation was documented in their hospital files by the clinical pharmacists.

### Study design

A randomized controlled study design was used to explore the study objective [16]. The included participants were randomized 1:4 (50:200) into a control and an intervention group

with parallel assignment using the randomization online system (randomization.com) by the Pharmacy's Clinical Trials Department; see the Data Analysis section for more details. The randomization was blinded and divided into block sizes of 5/10/15/20/25 with consecutive numbered, opaque sealed envelopes, that were subsequently distributed to the included participants. Patients were recruited in the hospital wards by the first three authors from MIC who were blinded to the allocation until the envelope was opened.

The control group (50 participants) received no intervention beyond standard care which does not include access to any information services post discharge. The intervention group (200 participants) were offered standard care and the opportunity to contact the MIH on all business days, should any medicine-related questions arise after hospital discharge. Upon enrollment, patients in the intervention group received a card with contact information for the MIH. All enquiries to the MIH were answered orally and/or in writing by either a pharmacist or a pharmacy technician from the MIC. All enquiries and the case handling were documented in the MIC's database and categorized into predefined types of categories, such as adverse effects, choice of therapy, dosage/administration, drug interactions and product information.

## Telephone interviews

A primary outcome of the present study was to evaluate whether the offer to contact the MIH influenced patient's concerns and perception of safety in relation to their medicines (S1 Appendix question 9–14) as compared to standard care. Further, patient satisfaction with the new MIH was included as a primary outcome (S1 Appendix, question 25). The primary outcomes were investigated through telephone interviews with the included participants, lasting on average between 20–30 minutes.

An interview guide was developed to assess patient satisfaction with the medicine information obtained during hospitalization, at discharge and given by the MIH. In order to investigate patients concerns and perception of safety in relation to medicines use, already validated questions from the Beliefs About Medicine Questionnaire proposed by Horne et al. [17] and from a study by Badiani et al. [6] were included in the interview guide. Further questions were developed by author one and nine and were included according to the study objectives and contextual setting; see S1 Appendix for the complete questionnaire. The guide was divided into three overall sections directed towards the participants in the control group, the intervention group and those who contacted the MIH. The interview guide was structured and contained both closed- and open-ended questions or statements the participants would rate using a five-point scale (1 = not at all and 5 = to a very great extent); see S1 Appendix. If a patient was not in any medical treatment prior to, during admission or after discharge, the patient was not able to provide an answer to all questions from the interview guide. Hence, the total number of respondents to each question may vary. Further, not all patients answered all questions, but the responses given were included in the study. The number of answers given to each specific question will be represented by "n", and the calculated percentages are based on the fraction of "n".

The questions in the interview guide were pilottested by 12 patients and 5 healthcare professionals, and the wording of some of the questions was subsequently optimized accordingly. All eligible participants were interviewed by telephone within 2–4 weeks after discharge to ensure that the patients remembered details about their hospital stay. The telephone interviews were performed by author one, two and three.

Participants were rated as non-responders if they declined to be interviewed or if they did not answer the phone within five attempts on five different days over a two-week period. In addition, participants were also rated as a non-responder if they showed cognitive or/and

communicative problems during the interview, making it difficult to complete the interview, or if the patient was readmitted or deceased.

## Clinical impact assessment

Another primary outcome of the present study was the clinical impact of MIH on patient safety. To assess the clinical impact, all enquiries were analyzed and categorized for MRPs and pharmacy interventions according to the Westerlund System; see Table 1 [18,19]. Furthermore, the impact on patient safety was assessed using the DOCUMENT system, where the clinical significance of each enquiry from the intervention group was determined individually by two clinical pharmacists each with approximately 10 years of experience within clinical pharmacy and medicine information (the first and fourth author) [20]. Any discrepancies were discussed in order to determine a concluding assessment of the clinical significance; see Table 1.

## Data analysis

No similar studies are available from which a sample size can be calculated.

This study used a randomized controlled study design, and it was set to detect a 10% improvement on a 5-point scale measuring patient satisfaction with medicine information and patient concerns about medicine. From the literature, a mean score of 3.5 ± 0.5 is anticipated [2,21–23]. The 10% was set by the authors as this was assessed to allow for detection of any relevant changes in the scores.

Power was set by the authors to 90% and p-values less than 0.05 were considered statistically significant. Thus, a sample size of 43 patients was calculated [24], but to adjust for dropouts, the sample size was set to 50 patients. It was estimated by the authors that approximately 20% of the patients in the intervention group would contact the MIH thus resulting in a (50:200) (control:intervention) allocation.

Quantitative data from the patient interviews were analyzed in Microsoft Excel (for Office 365 MSO) and IBM SPSS (Version 25) with descriptive statistics and simple parametric statistics. Mann-Whitney U test were used for analysis of ordinal variables (five-point scale) and Chi-Square test were used for nominal variables (gender, yes/no). Data are displayed as the mean of rating ± SD (Standard Deviation) unless otherwise specified. A confidence interval

**Table 1. Clinical impact assessment of MRPs and the pharmacy intervention.**

| |
|---|
| **MRPs classified using the Westerlund System [18,19]** |
| Uncertainty about aim of the medicine; adverse reaction; interaction; therapy failure; problem of administration; inappropriate time for the medication intake/wrong dosage interval; over-/underuse of the medicine, medicine duplication; inappropriate storage of the medicine; difficulty opening the medicine; other MRPs. |
| **Clinical significance of the pharmacy intervention described by the DOCUMENT system [20]** |
| **Nil:** No consequence to the patient |
| **Low:** Consequences to the patient are related to costs or information only |
| **Mild:** Consequences to the patient are that they have improved compliance or improved or prevented a minor symptom. The sign or symptom should not require a doctor's visit to treat |
| **Moderate:** When, if the intervention did not occur, it was likely that the patient would have had to visit the doctor because of the consequences. Also covers the situation where the pharmacist needs to refer the patient to the doctor because of the seriousness of the situation |
| **High:** When, if the intervention did not occur, it was likely that the patient would have<br>• had to go to a hospital because of the consequences. Also covers the situation where the pharmacist needs to refer the patient to a hospital because of the seriousness of the situation<br>• required assistance from a regular nurse visit, or would have required placement into residential care |

(CI) was calculated using the Clopper-Pearson method for the primary outcome data using nominal variables. An independent t-test was used to test for difference in age and number of medications by admission and discharge. Test for normality was done using Shapiro-Wilk test. If data did not show a normality distribution a Wilcoxon rank sum test was used to test for difference. Free text comments were not analyzed explicitly but were solely used to support the interpretation of the patient interview.

## Results

### Participants

A total of 946 patients were assessed for eligibility at the two hospital wards. Of these, 319 patients failed to meet the inclusion criteria and 377 patients either declined the invitation to participate, were asleep, discharged or transferred to another ward at the time of inclusion.

Thus, 250 participants were included in the study and randomized into a control group (n = 50) and an intervention group (n = 200). In the follow-up period, 98 participants (17 in the control group and 81 in the intervention group) were considered as non-responders. Finally, 33 participants were interviewed in the control group and 119 participants in the intervention group; see Fig 1.

The participants were included equally from the Emergency Department and from the Department of Respiratory Medicine, and their characteristics are presented in Table 2. No significant differences were found between control and intervention groups regarding gender, age or number of medications at admission or discharge respectively; see Table 2.

### Medicine information during hospitalization and after discharge

A total of 152 participants were interviewed by telephone and the interview guide in S1 Appendix was followed. For nominal data, participants answering either"yes","no","do not know" were included in the analysis, whereas ordinal data only included answers containing ratings from 1–5. Not all participants answered all the questions. Thus, in the following, the number of answers given to each specific question is represented by "n", and the results are provided with the calculated percentage based on the fraction of "n".

In the control group 55% of the participants (18/33) reported a change in their medication during hospitalization, see S2 Appendix. A similar level of change in medication was reported by participants in the intervention group (58%, 68/118), see S2 Appendix. Further, the participants expressed a high level of satisfaction (score >3) with the medicine information provided during hospitalization and at discharge, see Fig 2. No significant differences were seen between the control- and intervention group with regard to satisfaction with the medicine information provided during hospitalization or at discharge, see Fig 2.

Approximately half of the participants (48%, 73/141) preferred receiving their medicine information both written and orally at discharge, and 2/3 of these expressed that this combination gave them the opportunity to validate and repeat the obtained information when settled at home. Of the participants in the control and intervention group, 21% (7/33) and 29% (34/118) respectively did not recall that they received a medication-status-list at discharge. There was no significant difference between the groups (p = 0.377).

Of all participants, 26% (39/150) responded that they had questions regarding their medication after discharge (18% (6/33) in the control group and 28% (33/117) in the intervention group). The percentage of patients that had a question regarding their medication were higher in both control- and intervention group when looking at patients who reported a change in their medication (22% and 39,7% respectively). Among all patients who reported they had a question regarding their medicine a total of 67% in both the control group (4/6) and

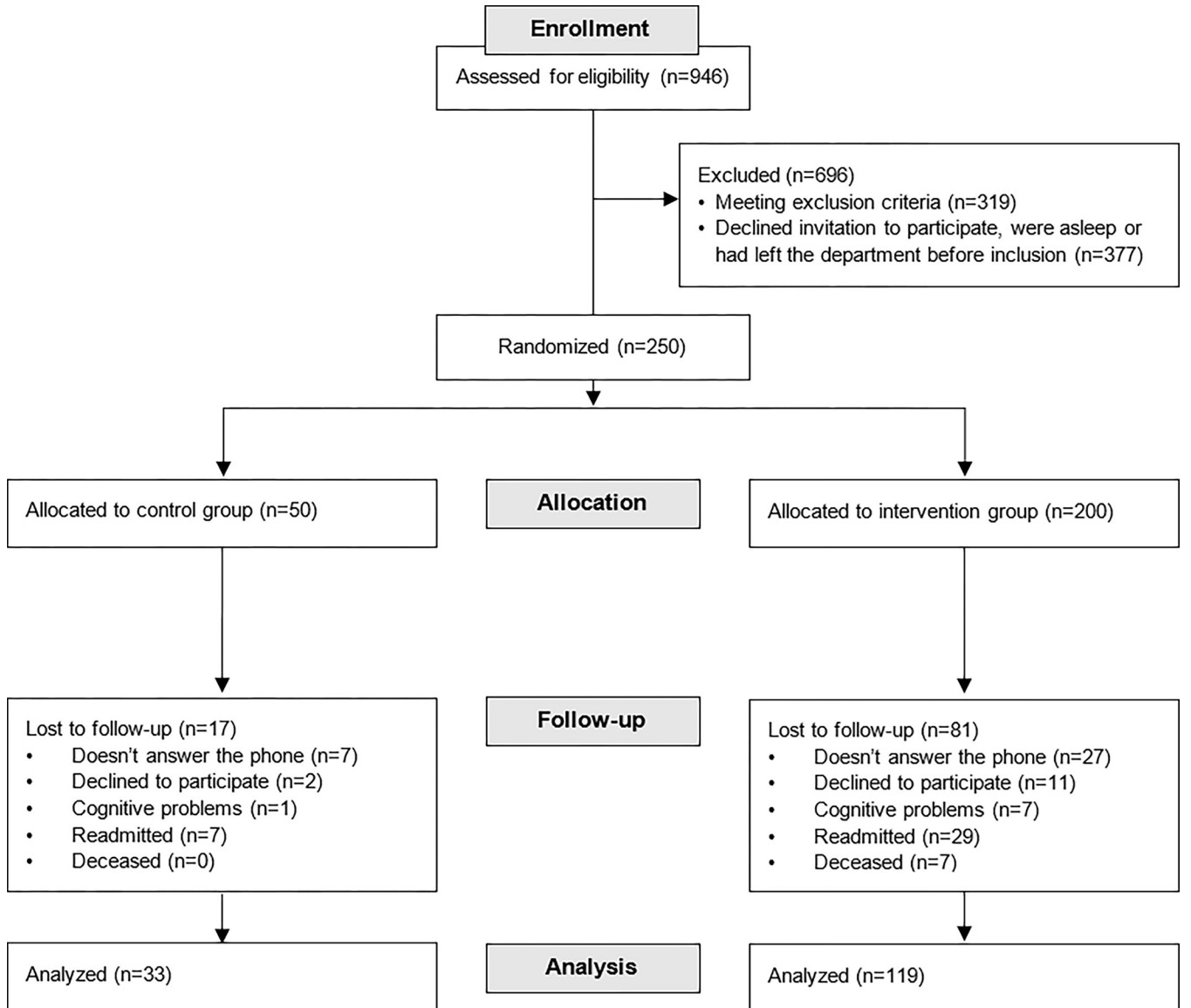

**Fig 1. Consort study diagram showing the flow of invited, included, excluded and dropout patients.**

intervention group (22/33) found an answer to their question by searching the internet or contacting their general practitioner, pharmacy, hospital, relatives or visiting nurses, see S2 Appendix. In total, 8 patients from the intervention group contacted the MIH before the telephone interview and 100% (8/8) of those received an answer to their question. Of these 8 patients 6 patients reported a change in their medication (75%).

## Patient satisfaction with MIH

One of the primary outcomes was patient satisfaction with MIH. Participants in the intervention group were asked if the opportunity to contact the MIH affected the feeling of safety in relation to their medication, which the majority of the participants acknowledged (72%, 79/110, CI [0.67–0.84[). Patients who contacted the MIH before the telephone interview (n = 8) reported a slightly lower score on the questions regarding feeling of safety with regard to their

**Table 2. Characteristics of enrolled participants.**

| Characteristics | Control group (n = 33) | Intervention group (n = 119) | All participants (n = 152) | P Value |
|---|---|---|---|---|
| Gender, No. (%)[a]<br>Male<br>Female | <br>13 (39)<br>20 (61) | <br>55 (46)<br>64 (54) | <br>68 (45)<br>84 (55) | 0.540 |
| Age, median (IQR) | 71 (57–78) | 67 (51–74) | 68 (54–75.5) | 0.143[C] |
| Hospital ward, No. (%)[a]<br>Acute medical ward<br>Department of respiratory medicine | <br>12 (36)<br>21 (64) | <br>60 (50)<br>59 (50) | <br>72 (47)<br>80 (53) | 0.152 |
| No. of medications by admission, median (IQR)[b] | 7 (5–12.5) | 8 (4–13) | 7,5 (4–12.5) | 0.766[C] |
| No. of medications by discharge, median (IQR)[b] | 9 (5.5–14.5) | 9 (5–14) | 9 (5–14) | 0.881[C] |

IQR: Interquartile range.

[a]Percentages of total number.

[b]Including all registered medicines in the hospital files at admission and discharge.

[C]Wilcoxon rank sum test.

medication and they felt to a higher degree that their medicines were a mystery to them (see S2 Appendix). Importantly, these patients expressed that the feeling of security increased after contacting the MIH, as they found the information reassuring and brought clarity to potential worries, such as side effects, dosage regimens etc. In addition, the patients reported that the helpline provided practical advice on how to administer the medication.

Only patients in contact with the MIH before the telephone interview were asked about their satisfaction with the service (n = 8). The participants contacted the MIH by phone and received an answer by phone. The participants found the answers provided by the MIH timely, easy to understand and relevant. Furthermore, they reported that the answers to a great extent affected their use of medicine in a positive manner, just as they were satisfied with the answers to a very great extent (4.71 ± 0.45 on a 5-point scale). Generally, the participants were satisfied with the service (4.57 ± 0.73 on a 5-point scale), they would use it again (100% answered yes, n = 8), and they believed the helpline should be a permanent service for discharged patients in the Capital Region of Denmark (93% answered yes, n = 126).

## Concerns and perception of safety in relation to medication

The other primary outcome focused on patients' concerns and perception of safety in relation to their medication. Both the control- and the intervention group responded that they felt safe

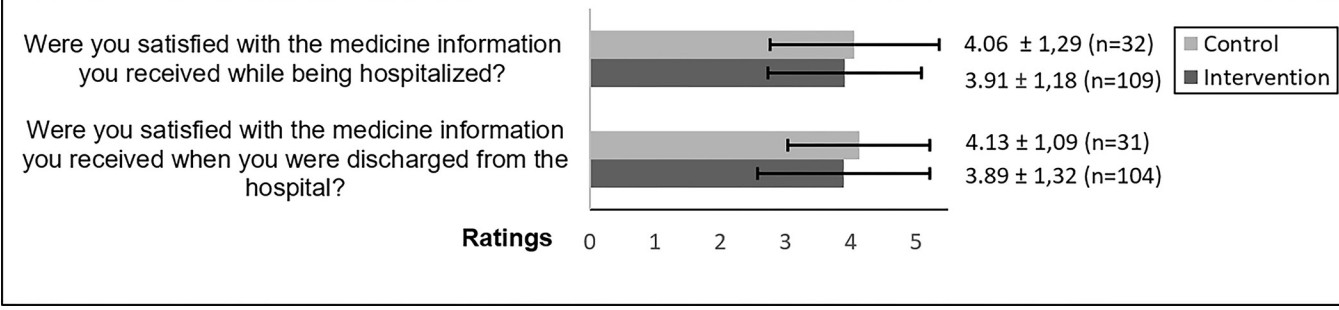

**Fig 2. Patient satisfaction with medicine information provided during hospitalization and after discharge.** Patients scored satisfaction with medicine information provided during hospitalization and at discharge on a 5-point scale where score 1 = not at all and score 5 = to a very great extent. Results are shown as mean of rating ± SD. No significant difference was found between the control- and intervention group with regard to satisfaction with medicine information provided during hospitalization (Mann-Whitney U test, p = 0.337) or at discharge (Mann-Whitney U test, p = 0.580).

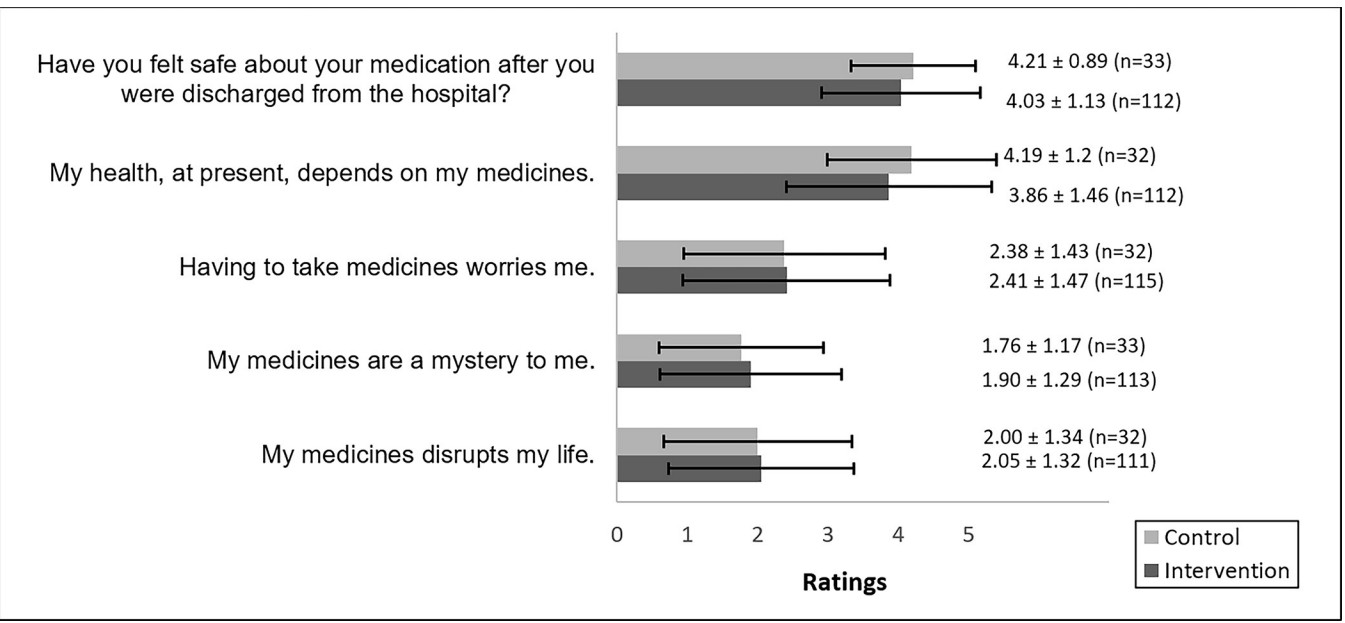

**Fig 3. Concerns and perception of safety about medicine.** Patients scored on a 5-point scale with 1 = not at all and 5 = to a very great extent. Results are shown as mean of rating ± SD. No significant difference was found between control- and intervention group for any of the questions (Mann-Whitney U test, p>0.05 for all questions).

to a great extent regarding their medication after discharge, which equals to a rating 4 or more on a 5-point Likert scale. There was no significant difference between the groups (p = 0.59). Both groups expressed that their current health to some extent or to a great extent was dependent on medication (no group difference, p = 0.33). However, they did not worry about taking their medicine (no group difference, p = 0.969), nor did they experience their medication as a mystery (no group difference, p = 0.507), and they expressed that the medication only disrupted their lives to a small extent (no group difference, p = 0.844) (see Fig 3).

## Enquiries to the MIH service

During the study period, the participants from the intervention group were offered the possibility to contact the MIH if they had any questions regarding their medication after discharge.

In all, 26 participants raised a total of 37 enquiries to the MIH pre-, post- or during the telephone interviews (S3 File). These enquiries were categorized into predefined types of categories. The majority of enquiries were related to dosage/administration and product information (effect and usage), see Table 3.

## Clinical impact of MIH on patient safety

MRPs and the clinical impact of the helpline responses were assessed by two clinical pharmacists working at the MIC. All enquiries were included in the assessment regardless of when the patient asked the question relatively to the telephone interview, i.e., pre-, post- or during the telephone interviews. From the 37 enquiries, a total of 43 MRPs were identified according to the Westerlund system (Fig 4). The most frequent MRP was "uncertainty about the aim of the medicine", e.g., uncertainty on how to take insulin or whether two prescribed medications during hospitalization were the same and should be administered similarly. Further, concerns about adverse reactions, interactions between medicines and therapy failure were identified MRPs in more than 10% of the enquiries (see Fig 4).

**Table 3. Categorization of enquiries to MIH from July 2018 through May 2019 (n = 37).**

| Category | Enquiries before telephone interview | Enquiries during telephone interview | Enquiries after telephone interview |
|---|---|---|---|
| Adverse effects, No. (%) | 1 (2.7) | 2 (5.4) | 3 (8.1) |
| Choice of therapy, No. (%) | 1 (2.7) | 0 (0.0) | 1 (2.7) |
| Dosage & administration, No. (%) | 3 (8.1) | 5 (13.5) | 3 (8.1) |
| Interactions, No. (%) | 1 (2.7) | 0 (0.0) | 3 (8.1) |
| Product information, effect and usage, No. (%) | 4 (10.8) | 6 (16.3) | 1 (2.7) |
| Other, No. (%) | 0 (0.0) | 1 (2.7) | 2 (5.4) |
| Total number | 10 (27%) | 14 (38%) | 13 (35%) |

MIH: Medicines Information Helpline. "Other" include enquiries about patients' control/treatment at the hospital and the need to speak with either a hospital- or general physician.

The pharmacy interventions for each MRP were also identified using the Westerlund system. One MRP may trigger several pharmaceutical interventions, and a total of 77 were identified. Of these, the pharmaceutical intervention *medicine counseling* was performed for all the 43 MRPs extracted from the 37 enquiries to the MIH. Further, the pharmaceutical interventions consisted of referral to prescriber or other health-care provider (17 situations); provision of practical information (12 situations); direct contact from the pharmacy personnel with the hospital prescriber (3 situations), and provision of information to patients' representative (2 situations), which adds up to a total of 77 pharmaceutical interventions.

Clinical significance was assessed using the DOCUMENT system [20]. However, in most cases the clinical significance could not be attributed to a single specific pharmacy intervention. Instead, it was a sum of several pharmacy interventions performed to alleviate the MRP. Thus, clinical assessment was performed at the level of each MRP. More than 50% of the MRPs resulted in pharmacy interventions assessed to have a high- or moderate clinical

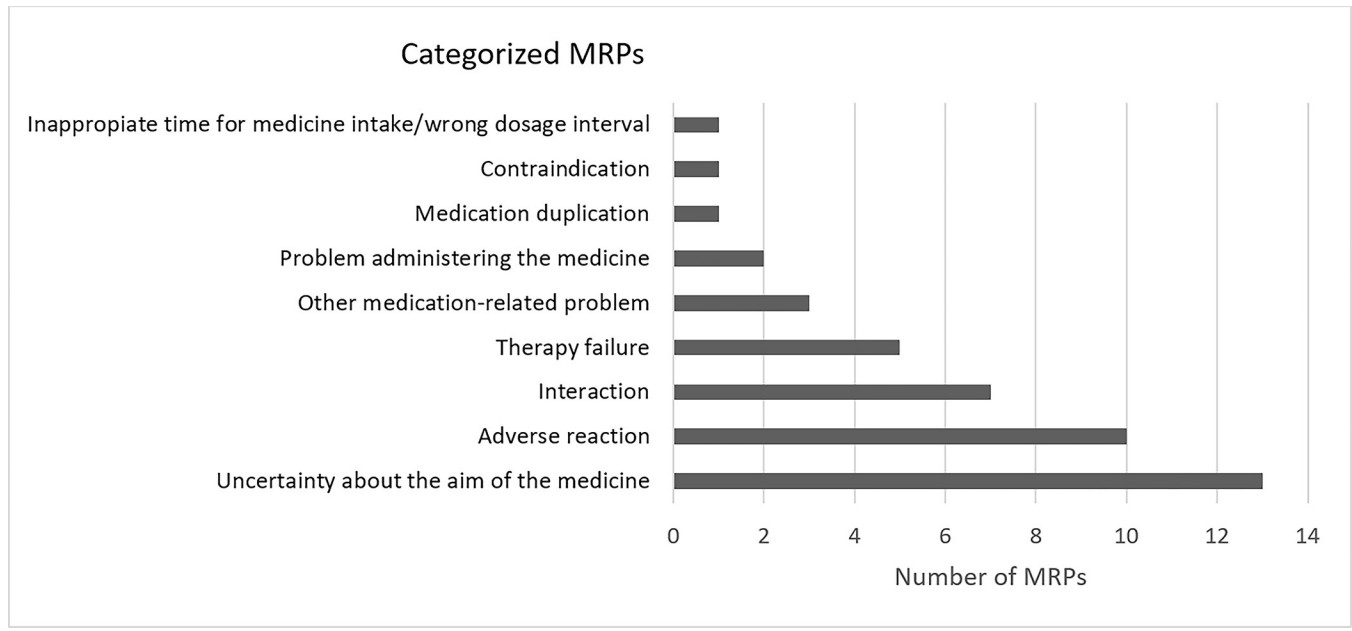

**Fig 4. Categorization of MRPs related to the enquiry according to the Westerlund system [18,19].** The 43 MRPs extracted from the MIH. Other MRPs included enquiries about vitamins and the need to speak with either a hospital- or general physician.

**Table 4. Assessment of the clinical significance of the pharmacy interventions proposed to mitigate MRPs.**

| Clinical significance | Number of MRPs (n (%)) (N = 43) |
|---|---|
| Nil | 1 (2) |
| Low | 8 (19) |
| Mild | 11 (26) |
| Moderate | 17 (39) |
| High | 6 (14) |

significance (see Table 4). Among these were: counselling on how to administer insulin; impact of concomitant treatment with ciprofloxacin and theophylline; and length of antibiotic treatment to a patient who had only received tablets for 3 days and no prescription.

## Discussion

The current study showed that patients discharged from hospital in general expressed a high level of satisfaction with the standard care medicine information provided by the health care personnel at the wards during hospitalization and at discharge. Despite this, a number of patients still had questions regarding their medication after hospital discharge. As expected, the rate of patients who had a question regarding their medicines was higher in the group of patients who reported having experienced a change in their medication during hospitalization indication that changes in medicine may lead to insecurity. Patients who were offered access to contact a newly established hospital-based MIH reported an increase in their feeling of safety in relation to their medication and all patients in contact with the MIH were "to a great extent" satisfied with the MIH service. However, asking all patients about their perception of safety in relation to medicines use revealed no difference between the control and the intervention group. During the study period, a total of 37 enquiries were received and answered. Analysis of the enquiries revealed 43 MRPs of which 53% triggered interventions assessed to have a moderate to high clinical significance.

### Need for medicine information after discharge

The present study shows that approximately one fourth of the participants (26%) express a needed for information about their medication after hospital discharge. Similar results have been reported by Shuen et al. who found that 26% of patients discharged from an emergency department made a call or visit to their primary medical doctor or a specialist physician post discharge [25]. In this study only 8 out of 200 patients contacted the MIH after discharge. However, performing the telephone interviews (post discharge) triggered another 18 patients to make a total of 37 enquiries. The fact that post discharge contact triggers medicine-related questions is not surprising, and several studies have explored the impact of different follow-up strategies such as post-discharge home visits by pharmacist, follow-up phone calls and text messages in order to alleviate MRPs [21,26,27]. Although the number of patients taking a non-stimulated contact to the MIH is limited in this study, it should be seen in the context that the service is newly established to patients. As the MIH becomes a well-known service, we anticipate that the number of enquiries increase over time. Indeed, it could be relevant to introduce follow-up contacts as this has been shown to be a very effective strategy in supporting patients in relation to their medication after discharge [21,26,27]. The present study did not explore barriers to contact the MIH and it could be relevant in future studies to elucidate patients' motivation and barriers to contact the MIH.

### Increased patient safety

Analysis of the patient enquiries to the MIH revealed that 53% triggered interventions assessed to have a moderate to high clinical significance, suggesting that the interventions improved the patient safety of the patient medication. It is well known that medication after discharge is indeed a major cause of rehospitalization [1–3]. Studies indicate that providing the appropriate level of medicine information after discharge may reduce medicine related rehospitalization or contact to primary medical doctor or physician [4,27,28]. In this context a MIH may not only improve the patient safety of the medication after discharge, but it may also decrease overall costs of treatment. However, further studies are needed to elucidate the cost-benefit of a MIH service to patients after discharge.

### High level of patient satisfaction with MIH

This study showed a high level of satisfaction with the MIH. The participants found the answers to their enquiries comprehensible and relevant, which affected their use of medicine. Other medicine helplines have reported similar high ratings of user satisfaction where the majority of the patients report that they received sufficient information and felt reassured about their medicine [8]. In line with the current study, Badiani et al. found that 95.9% of the respondents followed the advice provided in their study [6].

In this study, the most common enquiries were related to dose, administration, effect and usage of medicines. The nature of these enquiries is similar to those found in the literature [5,6,8]. Further assessment of each enquiry revealed 43 MRPs, where the most frequent MRP was uncertainty about the aim of the medicine, which indicates that the information given during hospitalization or at discharge was insufficient. Similar results have been reported from other medicine helplines where incorrect or insufficient information about medicine is the main topic of enquiries [5,8].

### Perception of safety with medicine after discharge

Patients in both groups reported that they to a high degree felt safe with their medication after discharge. Further, the current study revealed that patients with access to the MIH expressed an increase in the feeling of safety in relation to their medication; see S2 Appendix. Many patients take action and seek information from e.g., friends, family or the internet in order to prevent harm [29] and here MIHs may indeed be a relevant source of information helping the patients alleviating these worries and support and empower the patients to deal with the problems.

Relevant medical information not only enables the patients to prevent and solve MRPs, but it may also be a key factor in relation to medicine adherence [30]. In this study adherence was not directly evaluated, however, the increase in feeling of safety in relation to medication may have a positive effect on adherence. Future studies are needed to elucidate the role of MIH in relation to adherence.

### Methodological considerations

The present study was using a randomized study design. The dropout rate (enrolled patients not performing the telephone interview) in the current study was higher than anticipated (39%, 17 and 81 in the control and intervention group respectively) limiting the power to identify statistically significant differences. Applicable to both groups, the high dropout rate was mainly due to patients not answering their phone or being readmitted. For future studies, a suggestion to minimize the high dropout rate could be to schedule follow-up telephone calls or

to arrange a personal meeting in the patients' private homes and enroll patients who are less prone to be readmitted.

Another limitation in the current study concerns the interview guide that includes some questions that are not fully validated. However, most of the questions in the interview guide were adopted word-for-word from other studies ensuring that the majority of the questions had been tested by other researchers beforehand [6,17]. Furthermore, the final interview guide was pilot-tested, thus assuring an acceptable level of validity. The study did not include the full Beliefs About Medicine Questionnaire [17], thus not allowing a fully valid conclusion on patients concerns and the necessity of taking medicine. In this regard it should be considered if the sensitivity to measure the primary outcome regarding patients feeling of safety in relation to the medication was high enough.

The clinical impact of the helpline responses was assessed by two clinical pharmacists working at the MIC. This is a potential bias, and it would have strengthened the data analysis if an impartial party had analyzed the responses.

Another methodological consideration in the current study is the risk of respondent's social desirability bias where respondents may respond in a way they think the investigator wants to hear and hereby present themselves in the best possible way [31]. Social desirability bias may lead to a better rating of the medicine helpline service although neutral wordings were used in the questions and most questions were answered on a scale rather than single items.

Further, there is a risk of recall bias, which is a systematic error that occurs when the participants do not recall or remember previous experiences accurately or omit details related to the topic of interest [32,33]. Since the interview in this study took place 2–4 weeks after discharge, there is a risk of recall bias, especially with regards to the medicine information during hospitalization and at discharge. However, recall bias is considered more prominent in retrospective surveys or self-reports, and the methodological approach in this study was telephone interviews where the researchers had the opportunity to probe and clarify the questions during the interviews. Thus, the extent of recall bias is considered minimal. Another limitation aspect is the fact that 39% of the included patients were lost in the follow-up and thus they were never interviewed. Although a 61% response rate is considered acceptable a high dropout rate might affect the generalizability of the study results.

Participants in the current study were included from two departments, an emergency department and a department of respiratory medicine. Thus, the present results are most probably only representative of similar clinical settings. Indeed, future studies should investigate other clinical settings.

## Conclusion

MIH offers support for discharged patients to alleviate MRPs. More than 50% of the MRPs found in this study resulted in pharmacy interventions assessed to have a high- or moderate clinical significance. Although patients expressed a high satisfaction with the MIH, the access to the MIH did not significantly increase patients' feeling of safety in relation to medicines use. This study emphasizes that patients value the service and the information provided by the MIH, and that the service might positively affect a safe medication after discharge. Changes in the medical treatment after discharge is challenging and patients should be provided with sufficient pharmaceutical support and information to overcome these struggles, ultimately ensuring a patient safe medication.

## Supporting information

**S1 Checklist. CONSORT 2010 checklist of information to include when reporting a randomised trial*.**
(DOC)

**S1 Appendix. This is the interview guide.**
(DOCX)

**S2 Appendix. This is the results based in the interview guide.**
(DOCX)

**S1 File. This is the ethic committee protocol in Danish.**
(DOCX)

**S2 File. This is the ethic committee protocol in English.**
(DOCX)

**S3 File. This is the raw data from the case handling of request to MIH.**
(DOCX)

**S4 File. This shows means, medians and box plots for all data given in ordinal variables.**
(DOCX)

## Author Contributions

**Conceptualization:** Karianne Wilhelmsen Fjære, Tim Emil Vejborg, Lene Colberg, Lars Pedersen, Ann Kathrin Demény, Helle Byg Armandi, Marianne Hald Clemmensen.

**Data curation:** Karianne Wilhelmsen Fjære, Tim Emil Vejborg, Lene Colberg, Marianne Hald Clemmensen.

**Formal analysis:** Karianne Wilhelmsen Fjære, Tim Emil Vejborg, Lene Colberg, Cecilia Strøjer Ulrich, Joo Hanne Poulsen, Marianne Hald Clemmensen.

**Funding acquisition:** Marianne Hald Clemmensen.

**Investigation:** Karianne Wilhelmsen Fjære, Tim Emil Vejborg, Lene Colberg, Cecilia Strøjer Ulrich, Marianne Hald Clemmensen.

**Methodology:** Karianne Wilhelmsen Fjære, Lars Pedersen, Ann Kathrin Demény, Marianne Hald Clemmensen.

**Project administration:** Karianne Wilhelmsen Fjære, Tim Emil Vejborg, Lene Colberg, Cecilia Strøjer Ulrich, Joo Hanne Poulsen, Helle Byg Armandi, Marianne Hald Clemmensen.

**Resources:** Marianne Hald Clemmensen.

**Supervision:** Lars Pedersen, Ann Kathrin Demény, Helle Byg Armandi, Marianne Hald Clemmensen.

**Visualization:** Karianne Wilhelmsen Fjære, Tim Emil Vejborg, Lene Colberg, Joo Hanne Poulsen, Marianne Hald Clemmensen.

**Writing – original draft:** Joo Hanne Poulsen, Marianne Hald Clemmensen.

**Writing – review & editing:** Karianne Wilhelmsen Fjære, Lene Colberg, Cecilia Strøjer Ulrich, Lars Pedersen, Ann Kathrin Demény, Helle Byg Armandi.

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
