## [Decision Letter · Decision Letter 0]

6 Mar 2023

PONE-D-22-33407Medicine Information Helpline after hospitalization – a randomized trial: impact on patient satisfaction, patient belief in medicine and clinical outcome on patient safetyPLOS ONE

Dear Dr. Clemmensen,

Thank you for submitting your manuscript to PLOS ONE. After careful consideration, we feel that it has merit but does not fully meet PLOS ONE’s publication criteria as it currently stands. Therefore, we invite you to submit a revised version of the manuscript that addresses the points raised during the review process. The manuscript has been evaluated by four reviewers, and their comments are available below. The reviewers have raised concerns regarding the reporting, methodology and statistical analysis of this study. Furthermore, the reviewers feel that the study’s limitations need to be discussed in more detail. 

Could you please revise the manuscript to carefully address the concerns raised?

We look forward to receiving your revised manuscript.

Kind regards,

Johannes Stortz, PhD

Staff Editor

PLOS ONE

Journal Requirements:

2. Thank you for submitting your clinical trial to PLOS ONE and for providing the name of the registry and the registration number. The information in the registry entry suggests that your trial was registered after patient recruitment began. PLOS ONE strongly encourages authors to register all trials before recruiting the first participant in a study. As per the journal’s editorial policy, please include in the Methods section of your paper: 1) your reasons for your delay in registering this study (after enrolment of participants started); 2) confirmation that all related trials are registered by stating: “The authors confirm that all ongoing and related trials for this drug/intervention are registered”.

Reviewers' comments:

Reviewer's Responses to Questions

**Comments to the Author**

1. Is the manuscript technically sound, and do the data support the conclusions?

Reviewer #1: Partly

Reviewer #2: Partly

Reviewer #3: Partly

Reviewer #4: Yes

2. Has the statistical analysis been performed appropriately and rigorously? 

Reviewer #1: Yes

Reviewer #2: No

Reviewer #3: Yes

Reviewer #4: Yes

3. Have the authors made all data underlying the findings in their manuscript fully available?

Reviewer #1: No

Reviewer #2: No

Reviewer #3: Yes

Reviewer #4: Yes

4. Is the manuscript presented in an intelligible fashion and written in standard English?

Reviewer #1: Yes

Reviewer #2: Yes

Reviewer #3: Yes

Reviewer #4: Yes

5. Review Comments to the Author

Reviewer #1: Thank you for the opportunity to look at your work. I provided my individual comments at relevant sections. I didn't look at the references and conclusion. Also suggest to double check the references. Please refer to attached pdf file. Hope they are useful.

Reviewer #2: The randomized-controlled clinical trial evaluated discharged patient satisfaction with a newly established Danish hospital-based medicine information helpline (MIH) by describing the service’s effects on patient beliefs and perception of security about their medical treatment. The study also assessed the clinical impact of MIH on patient safety. The conclusions are unclear.

Major revisions:

1- Abstract: Briefly state the statistical method used from which the conclusions were drawn and provide p-values to support the statements.

2- If age is normally distributed, provide means and standard deviations as summary measures and compare groups using an independent t-test. If age is non-normally distributed, summarize with medians, first and third quartiles and compare groups using the Wilcoxon rank sum test. Use the Shapiro-Wilk test to test if the distribution of data deviates from a comparable normal distribution.

3- Apply the same approach as stated in comment 4 to the number of medications at admission and discharge.

4- Line 228: Provide statistical justification for the following statement. “No significant differences were seen between the control- and intervention group.”

5- Express p-values more precisely than p>0.05.

6- Line 254: Provide statistical justification for the following statement. “No significant difference was found between the groups.”

7- Line 272: State the statistical methods used to estimate the values of 4.17 and 0.48.

8- Figures: Determine if these variables are normally distributed. Perhaps a box plot is more appropriate if the data does not follow a normal distribution. In this case, summarize the variables using median, first and third quartiles.

9- The drop-out rate was extremely high. Discuss this limitation.

Minor revisions:

1- Line 163 contains a grammatical error.

2- Indicate the date range subjects were enrolled in the study.

Reviewer #3: This manuscript describes a randomised controlled trial of a medicines information helpline (MIH) designed to support patients recently discharged from hospital. 250 participants were randomised to have access to a MIH, or not to have access to a MIH. Patient reported outcomes on medicines information needs, beliefs about the security of medicines and helpline satisfaction were collected from 152 participants. Of the 119 of these participants randomised to MIH access, only 8 contacted the helpline before data collection.

There were no significant differences between the two randomised groups, but 72% of participants in the MIH group reported that the availability of the MIH made them feel safer. The MIH also identified and potentially resolved 43 medicines related problems (MRPs), 23 of which had moderate or high clinical significance.

MIHs are widely used in some countries, but there is limited and weak evidence of their effectiveness. A randomised study of an MIH is therefore necessary, making this study a novel contribution to the literature, which may be interesting and useful to readers of PLoS ONE.

The authors clearly faced challenges in delivering the study, as 98 participants were lost to follow up. In addition, only 8 participants had contacted the MIH before outcome data were collected, meaning the study is underpowered to detect a difference between the two groups. I think this has affected how the authors have reported their results, as the Results and Discussion sections focus on user satisfaction and MRPs from the MIH, and do not emphasise the comparison of outcomes in the two groups as would normally be expected in a RCT report. I therefore suggest the following recommendations.

Major comments:

1) State the primary and secondary outcomes more specifically. The primary outcomes are described in lines 127-130 as evaluating perceptions of security about medicines use and satisfaction. However, these constructs relate to 5 and 14 questions, respectively, in the follow up survey. So, I suggest it is clearly stated which of these questions were considered the primary outcomes.

2) Include these primary outcomes earlier in the results section, immediately after participant characteristics are described. Clearly describe them as the primary outcome and consider including an estimated effect size and 95% confidence interval for each one (as per CONSORT 2010). A description of the secondary outcomes can then follow.

3) Be clear at the start of the discussion and in the conclusion that no effect of the MIH was seen on the primary outcome, as 39% of participants were lost to follow up and there were not many calls to the helpline – both of which are major limitations. Consider discussing what the authors learned from this study that might be useful for future authors attempting a similar project.

4) In a similar way, make sure the primary outcomes are clearly stated in both the Methods and Results sub-sections of the abstract, and include numerical data in the results.

Minor comments:

Lines 111-125: as per CONSORT 2010, state who randomised the participants, as well as who recruited them.

Lines 111-125: as per CONSORT 2010 state whether the investigators and statisticians were blinded to the allocation. For clarity, state explicitly if the study was completely open label.

Line 138: the interview guide is described as ‘semi-structured’. This implies a qualitative method, with the guide used flexibly in response to what participants said – which would be inappropriate for an RCT. However, the interview guide in Appendix S2 appears to be a ‘structured’ (not semi-structured), so I would change this description.

Lines 168-169: include a reference for the type of calculation used to determine sample size.

Lines 162-176: how were ‘free text’ responses analysed?

Lines 162-176: clearly state whether or not the data were analysed on an intention to treat basis.

Lines 175-176: Differences in age and number of medications are analysed using the t-test, yet these results are summarised as the median and IQR in Table 2. This implies that they were not normally distributed, in which case the Mann-Whitney U test would be more appropriate.

Line 277: Clinical impact was assessed by pharmacists providing the MIH. This is a potential source of bias, which I suggest is discussed in the limitations section.

Figure 1: consider extending this to include how many participants called the MIH before, during and after the interviews

Figures 2&3 (and the sections of Appendix S3 that report the same data): define what values are reported for each Likert scale outcome. I assume it’s mean +/- standard deviation, but this is not stated.

There are multiple minor English language problems – I suggest a thorough proof-read by an experienced writer in English.

Reviewer #4: Thank you for a great initiative to improve the care transition concerning patients’ need for more/better medication information. I believe the approach is very appealing; however, I think your conclusion should be slightly more moderate. This includes updating the abstract. Further, I have some comments about potential sub-analyses to understand your data better, which you should include if possible. I also think you need to compare your intervention more to other similar ones and discuss how you might want to develop/modify yours. For example, only 8 patients contacted you without a reminder, maybe patients need some sort of reminder.

Titel

Medication instead of medicine? I’d recommend medications in more formal language and medicines in more lay language.

Patients’ beliefs about medicines/medications?

Abstract

Who performed the MIH service? Nurses, pharmacists?

Line:35/36 affects the patient’s belief and perception of security about their medical treatment= referring back to the title do you mean beliefs about medical treatment of only medications specifically?

43 helpline, patient belief about medicine. Add (BMQ)

43 helpline, patient belief about medicine-> patient’s beliefs

Line 44/45 Medication-related problems (MRPs) were identified and clinical impact of the MIH service was assessed. Of the person performing the service or during the telephone interviews?

47/48: thirty-seven questions were enquired by 26 participants to the MIH during the study-> please also add the percentage of how many had used the service and numbers from pre and post-trigger as well.

they all?n=26?

Line 48:Participants in the intervention group reported that the MIH increased their sense of security regarding medicine= all 119?

From line 249: Both the control- and the intervention group responded that they felt safe to a great extent regarding their medication after discharge, and there was no significant difference between groups (p=0.59).

52 Conclusion: The MIH was highly appreciated by the participants, which indicate that the MIH is a valuable service for discharged patients in improving the sense of security in relation to medication and alleviating MRPs.

Please make this statement a little more modest based on the results from line 249.

Background

58: Transition from hospitals to private homes is known to challenge (the) patient safety. Remove the, and please add a reference.

medication-related problems (MRPs), why MRPs and not DRP? How do you define the concept? You use DRPs in the conclusion.

Line 72: Medicine Information Helplines (MIH) can play an important role in supporting patients in a safe medication after hospitalization. Add a reference.

74: A pharmacy professional. Wording, pharmacist?

Do other countries use MIH or only UK?

80:Patient may enquire the general practitioners or community pharmacies with medicine-related questions [9]. not the hospital?

85: This study is to our knowledge the first to explore if addition of MIH affects patient’s beliefs and perception of security about medicines as compared to standard care with limited medicine information support after discharge. Is this totally new or only in the Danish context? Please clarify.

I lack a summary of other possible services also aiming at supporting patients after discharge in regard to medication use. What makes MIH an alternative above other possible options?

Method

Please start with the study design first.

Setting->? Description of the MIH?

Please describe the service in more in-depth. How many pharmacists offer the services, how were they trained in e.g. telephone consultations skills? Opening hours. Did some of them always carry a phone with them? Did you pilot the service first?

Also briefly describe the hospital, its size?

101: were recruited in the hospital wards-> in the two selected?

Were it some inclusion criteria regarding the length of the stay at the ward? Did they have to use medications? Did they have to have had a medication change during the stay as this probably will impact their perceived information needs?

Study design

112 A randomized controlled study design was used to explore the study objective. Reference to a method book?

The included participants were randomized 1:4 (50:200). Why this ratio? Add a reference. I see you discussed this in the analysis section, maybe refer to that section?

123 pharmacy technician from the MIC. Potential questions were documented. Only questions no actions/responses?

135. and from a study by Badiani et 135 al. (2017) [7] were included in the interview guide. Is that something different from Hornes? Please describe. Add also what Danish version you used.

136 further questions were included according to 136 the study objectives and contextual setting. Such as? Or refer to the appendix earlier.

All eligible participants were interviewed by telephone within 145 2-4 weeks after discharge. How did you decide upon the timeframe? Add an argument.

140: However, if a patient is not in any medical treatment prior to, during admission or after discharge, the patient is not able to provide an answer to each question from the interview guide.

With medical treatment? Do you then mean medication use? If yes, please perform a sub-analysis with these persons excluded as this probably impacts your user rate of the service. I.e. do you have some data on which patients had some changes in their medication use, it would be interesting to see if these people more frequently used your service. If possible add this information, if not possible include it in the method discussion.

156 enquiry from the intervention group was determined individually by two clinical pharmacists. Why didn’t you include a medical doctor as well in this assessment?

162 No similar studies are available from which a sample size can be calculated. What about the studies from UK?

Results

How long were the consultations on average?

208 Enquiries to the MIH->MIH service

Table 3

Explain the abbreviation MIH under the table.

Enquiries before

interview

Enquiries during

interview

Enquiries after

Interview

I don’t understand the categorization, do you mean during the consultation? What do you mean by after? Please add this information/categorization to your method section as well.

Enquiries during interview. Can you count this? I think you need to add these pre-post data in e.g. your abstract.

In the control group 55% of the participants (18/33) reportedly experienced a change in their medication during hospitalization, see S3 appendix. A similar level of change in medication was reported by participants in the intervention group (58%, 68/118), see S3 appendix. See previous comment. I’d like you to perform a subanalysis to see how this impacts the use of the service.

Line 233: Different formatting of the text

233 Patients scored satisfaction, with what?

228 hospitalization and at discharge (figure 2).-> Figure 2.

236 Approximately half of the participants (48 %, 73/141) preferred receiving their medicine information both written and orally. At discharge?

244:Among those, 67 % in both the control group (4/6) and intervention group (22/33) found an answer to their question by searching the internet or contacting their general practitioner, pharmacy, hospital, relatives or visiting nurses. Please refer to Appendix 3 for the distribution? N=X pharmacy.

246: in total, 8 patients from the intervention group contacted the 247 MIH and 100% (8/8). Didn’t you state: line 48: Thirty-seven questions were enquired by 26 participants to the MIH during the study,

263:Further, patients who contacted the MIH reported that the feeling of security increased after contacting the MIH, as they found the information reassuring and brought clarity to potential worries, such as side effects, dosage regimens etc. In addition, the patients informed that the helpline provided practical advice on how to administer the medication.

The patients contacting the MIH what characteristics did they have? What kind of beliefs about medicines did they have, did they differ from the others?

267 Only patients in contact with the MIH before the telephone interview were asked about their satisfaction with the service (n = 8).?? I don’t understand, please elaborate on this. Why didn’t you ask the otherones?

The pharmacy interventions? What do you mean here?

314 Despite this a considerable number of patients still had questions regarding their medication after hospital discharge. Do you refer to the Appendix 3 question. I don’t agree as if I understand right only about 20% contacted the MIH 26/119=21%

. Do you know why not more contacted your service of the 28% having questions?

Discussion.

I lack a discussion about how you would like to improve the MIH offered and a comparison to other similar services. In general, what outcomes do you think are important to assess? If someone wants to perform a similar service that would be of interest.

316/317: Patients who were offered access to the helpline reported an increase in their feeling of security in relation to their medication. Yes, but the groups did not differ regards beliefs about medicines etc. Please add this information.

323 The present study shows that patients have a need for information about their medication after discharge, more specifically, 26 % of the participants in the current study. 26/119?=21%? Or which numbers do you refer to? Is it question 6?

333 MIH is limited in this study, it should be seen in the context that the service is newly established to patients. As the MIH becomes a well-known service, we anticipate that the number of enquiries increase over time. So, you don’t think you need some more personal follow up to increase the numbers? Do you have some data from the patients on this what might make them contact MIH?

340: Studies indicate that providing the appropriate level of medicine information after discharge may reduce medicine related rehospitalization or contact to primary medical doctor or physician [1, 4, 23, 24]. In this context a MIH may not only improve the patient safety of the medication after discharge but it may also decrease overall costs of treatment. However, further studies are needed to elucidate the cost-benefit of a MIH service to patients after discharge. Do you have some data on how many also had tried to contact their doctor and you parallel?

350 with the current study, Badiani et al. found that 93% of the respondents followed the advice provided in their study [7]. How do you know this that they followed your advice?

361 Although patients in both groups reported that they to a high degree felt safe with their medication after discharge, the current study revealed that patients with access to the MIH expressed an increase in the feeling of security in relation to their medication. Do your data support this claim? Please reframe.

379: The study did not include the full Beliefs About Medicine Questionnaire [14], thus not allowing a fully valid conclusion on patients concerns and necessity of taking medicine. Why did you not include that?

383 A methodological consideration in the current study is the risk of respondent’s social desirability. Who did perform the interviews?

Other important outcomes to assess in a future study in addition to adherence?

Conclusion

405 MIH offers an important support for discharged patients to alleviate DRPs. Here you use DRPs not MRPs please be consistent and that the service positively affects a safe medication after discharge. Might impact.

Please make sure you answer your aims in the conclusion. Also add information about how many used the service pre/post prompt.

405 MIH offers an important support for discharged patients to alleviate DRPs and increase the belief and perception of security with their medicine after discharge. Can you draw that conclusion? The groups appear to have similar beliefs about medicines and level of security.

6. PLOS authors have the option to publish the peer review history of their article (what does this mean?). If published, this will include your full peer review and any attached files.

Reviewer #1: **Yes: **Gereltuya Dorj

Reviewer #2: No

Reviewer #3: No

Reviewer #4: No

---

## [Author Response · Author response to Decision Letter 0]

17 Jun 2023

Journal requirements 

1: Thank you for underlining the importance of the format. We have revised the manuscript according to the standards provided and hope it fulfills your requirements. 

2: The authors apologize for this mistake in not registering the study before starting the inclusion of the first patients. We have added a comment as requested to the manuscript.

3: We have revisited the raw data of the present manuscript. The data have been anonymized, so they comply with the informed consent as well as GDPR regulations. As requested, the data have been submitted as S6 Supporting Information. 

4: Captions for the supporting Information files have been included at the end of the manuscript as requested. 

Reviewer 1: 

Line 40: Thank you for the comment. The manuscript has been revised as suggested. 

Line 52: We are unfortunately not able to give an exact number as this statement relates to more than one question – ex question 14, 22, 25 all combined to an overall evaluation about patient satisfaction from the patient-telephone interview. 

Line 80: Thank you for the comment. The manuscript has been revised as suggested. 

Line 92: Thank you for the comment. The manuscript has been revised as suggested. 

Line 102 – see line 99, 100 and 101: Thank you for the comment. We have aligned the wording as suggested.

Line 104: We have aligned wording as suggested. 

Line 130: Thank you for the comment. The manuscript has been revised as suggested. 

Line 136: Than you for the comment. A clarification has been added to the manuscript. 

Line 144: We have emphasized that revision after pilot testing was revision of wording. 

Line 147: Thank you for the comment. We have added a clarification and revised the manuscript as suggested. 

Line 150: A detailed description of the inclusion of responders are given in figure 1. We did not experience a postponement of the interview when we first got in contact with a respondent. 

Line 157: Thank you for the comment. A clarification of the experience level has been added to the manuscript 

Line 192: We are not sure about this comment. Please provide a clarification if further revision is deemed. 

Line 206: Thank you for the comment. A clarification has been added to the manuscript. 

Line 208: We are not sure what is meant by “validated separately”. All questions were received separately but some contacted the MIH more than once. 

Line 211: A clarification has been added to the manuscript.

Line 214: The manuscript has been updated as suggested.

Line 223: Thank you for the comment. The manuscript has been revised as suggested. 

Line 224: If there was a change in medicine, the patient stated it during the interview, but no details on change on dosage, brand etc. was requested. This was regarded as face value. All raw data are now provided as supplementary material. 

Line 227: Thank you for the comment. We believe that the statement is supported by the data shown in figure 2. We have added a clarification that we make the conclusion based on a score >3. 

Line 241: The manuscript has been updated as suggested by another reviewer. 

Line 249: Thank you for the comment. The manuscript has been revised as suggested. 

Line 250 : Thank you for the comments. The manuscript has been updated accordingly. 

Line 256 + 262: We believe Safety is the correct term but that they can be used interchangeably. We have updated and aligned the wording. 

Line 272: Thank you for the comments. Quantified data has been added to the manuscript as suggested. 

Line 289: We understand the confusion and have suggested to delete the sentence as it does not really add any value to the current data. 

Line 293 + line 295: Thank you for the comments. We have tried to clarify that all MRP included “medicine counseling” and then further detail how many other interventions that were provided for the 43 MRPs. 

Line 293: We have tried to clarify that the listed interventions add adds up to a total of 77 pharmaceutical interventions.

Line 321: Thank you for the comments. We agree that the discussion may not be clear and we have updated the manuscript to specify and make it more clear to the reader. 

Line 324: We thanks for the comment and have revised the manuscript as suggested. 

Line 327: We thank you for this comment. We would like to emphasize that half of the patients enrolled in the current study are ED patients and we thus believe that the reference is valid in this context. 

Line 350: We apologize for this mistake and have revised the manuscript as requested. 

Line 360: We believe Safety is the correct term and have thus kept the wording. 

Line 374: The 60 % is the number of patients that were contacted but where enrollment was not succeeded. The 39 % is the dropout rate for respondents that were actually enrolled but did not perform the – please see figure 1. We have tried to clarify this further in the manuscript. 

Line 376: Thank you for the comments. We have deleted the adjective “smaller”. We believe however, that the pilot testing of the interview guide does account for a minimum level of validity. 

Line 379: We agree with the reviewers point and the consideration is taken for future studies.

Line 395: A revision of the manuscript has been provided earlier and we hope that the dropout rate is now more clearly explained and discussed. 

Line 398: We thank you for the comments which is taken to consideration for future studies

Reviewer 2: 

Major revisions

1- Abstract: Briefly state the statistical method used from which the conclusions were drawn and provide p-values to support the statements.

We thank you for the comment. We have tried to update the abstract accordingly so both statistical method and p-values are clear from the abstract. However, one of the primary endpoints consist of four questions where we have only provided the overall statistics in the abstract. 

2- If age is normally distributed, provide means and standard deviations as summary measures and compare groups using an independent t-test. If age is non-normally distributed, summarize with medians, first and third quartiles and compare groups using the Wilcoxon rank sum test. Use the Shapiro-Wilk test to test if the distribution of data deviates from a comparable normal distribution.

The authors thank for the comment and has revised the manuscript accordingly. 

3- Apply the same approach as stated in comment 4 to the number of medications at admission and discharge.

The authors thank for the comment and has revised the manuscript accordingly and table 2 have been updated with statistical clarification. 

4- Line 228: Provide statistical justification for the following statement. “No significant differences were seen between the control- and intervention group.”

A clarification has been added to the manuscript and further statistical justification and clarification has been provided in the legends to figure 2.

5- Express p-values more precisely than p>0.05.

A full revision of the manuscript has been done and all exact p-values are now provided in the manuscript. 

6- Line 254: Provide statistical justification for the following statement. “No significant difference was found between the groups.”

A clarification has been added to the manuscript and further statistical justification and clarification has been provided.

7- Line 272: State the statistical methods used to estimate the values of 4.17 and 0.48.

The values provided in line 272 are the mean of rating ± SD. A clarification has been added to the methods. 

8- Figures: Determine if these variables are normally distributed. Perhaps a box plot is more appropriate if the data does not follow a normal distribution. In this case, summarize the variables using median, first and third quartiles.

A test for normality has been applied and the statistics been adjusted accordingly where relevant. As this manuscript is not based on hard core qualitative data non-parametric tests have been applied throughout the manuscript. Levene’s test for equal varians has been applied for all ordinal variables (five-point scale) and no differences was found. We believe that showing the means gives a more honest presentation of the data. However, we do agree that box plots can add to the understanding of the raw data. Thus, box plots and medians for all questions that given in ordinal variables (five-point scale) have been included in S7 Supporting Information.

9- The drop-out rate was extremely high. Discuss this limitation.

Further clarification of the drop-out rate has been added to the manuscript and the limitation has been further discussed. 

Minor revisions:

1: We apologize for the mistake and the error has been corrected. 

2: The date range is already included in the study in line 100. 

Reviewer 3: 

1: State the primary and secondary outcomes more specifically. The primary outcomes are described in lines 127-130 as evaluating perceptions of security about medicines use and satisfaction. However, these constructs relate to 5 and 14 questions, respectively, in the follow up survey. So, I suggest it is clearly stated which of these questions were considered the primary outcomes.

Thank you for this relevant input. The questions considered as primary outcomes have been specified in the manuscript as suggested. 

2: Include these primary outcomes earlier in the results section, immediately after participant characteristics are described. Clearly describe them as the primary outcome and consider including an estimated effect size and 95% confidence interval for each one (as per CONSORT 2010). A description of the secondary outcomes can then follow.

We thank the reviewer for this suggestion. After careful consideration we believe that the logic way of presenting data follow the patient journey. Thus, we start with the medicine information provided during hospitalization and at discharge. We have then rearranged the section moving to patient satisfaction with the MIH before we go to the primary outcome regarding believes and perception of safety in relation to medication. We hope that this rearrangement of the data makes the reading logic and easy to follow.

3: Be clear at the start of the discussion and in the conclusion that no effect of the MIH was seen on the primary outcome, as 39% of participants were lost to follow up and there were not many calls to the helpline – both of which are major limitations. Consider discussing what the authors learned from this study that might be useful for future authors attempting a similar project.

A clarification has been added to both the start of the discussion and the conclusion. Further discussion of the drop out rate have been added. The authors believe that another limitation could be the sensitivity of the questionnaire and a small discussion have been added on this issue, 

4: In a similar way, make sure the primary outcomes are clearly stated in both the Methods and Results sub-sections of the abstract, and include numerical data in the results.

We thank for the input hand have tried to clarify throughout the manuscript when primary outcome data are described and discussed. 

Minor comments:

Lines 111-125: as per CONSORT 2010, state who randomised the participants, as well as who recruited them.

We thank for the comment and have added information on who randomized the patients and who recruited them.

Lines 111-125: as per CONSORT 2010 state whether the investigators and statisticians were blinded to the allocation. For clarity, state explicitly if the study was completely open label.

A description of the blinding of the investigator has been added to the manuscript. We do not understand the relevance of the blinding of the statistician as no analyses were performed before inclusion of all data was final.

Line 138: the interview guide is described as ‘semi-structured’. This implies a qualitative method, with the guide used flexibly in response to what participants said – which would be inappropriate for an RCT. However, the interview guide in Appendix S2 appears to be a ‘structured’ (not semi-structured), so I would change this description.

We thank for the comment and agree with the reviewer. The guide was followed as displayed in Appendix S2. We have revised the manuscript accordingly. 

Lines 168-169: include a reference for the type of calculation used to determine sample size.

A reference has been provided as requested. 

Lines 162-176: how were ‘free text’ responses analysed?

Free text comments were not analyzed explicitly, but were solely used to support the interpretation of the patient interview. This information has been added to the manuscript.

Lines 162-176: clearly state whether or not the data were analysed on an intention to treat basis.

The authors are not sure if this statement makes the analysis much clearer. It is clearly stated throughout the paper that patients were excluded for several reasons and thus not included in the analysis as would have been the case if an ITT approach was used.

Lines 175-176: Differences in age and number of medications are analysed using the t-test, yet these results are summarised as the median and IQR in Table 2. This implies that they were not normally distributed, in which case the Mann-Whitney U test would be more appropriate.

The authours agree and have revised the statistics regarding age and number of number of medication using a non-parametric test proposed by another reviewer.

Line 277: Clinical impact was assessed by pharmacists providing the MIH. This is a potential source of bias, which I suggest is discussed in the limitations section.

The authors agree and a short discussion of this limitation have been added to the manuscript. 

Figure 1: consider extending this to include how many participants called the MIH before, during and after the interviews

We believe that the data requested by the reviewer is available I table 3. 

Figures 2&3 (and the sections of Appendix S3 that report the same data): define what values are reported for each Likert scale outcome. I assume it’s mean +/- standard deviation, but this is not stated.

We apologize for the confusion and have provided the information as requested. 

Reviewer 4:

Reviewer #4: Thank you for a great initiative to improve the care transition concerning patients’ need for more/better medication information. I believe the approach is very appealing; however, I think your conclusion should be slightly more moderate. This includes updating the abstract. Further, I have some comments about potential sub-analyses to understand your data better, which you should include if possible. I also think you need to compare your intervention more to other similar ones and discuss how you might want to develop/modify yours. For example, only 8 patients contacted you without a reminder, maybe patients need some sort of reminder.

Titel

Medication instead of medicine? I’d recommend medications in more formal language and medicines in more lay language.

Patients’ beliefs about medicines/medications?

Thank you for the comment. We have updated the title but have keept “medicines” as this is also the word used in the original title for the BMQ.

Abstract

Who performed the MIH service? Nurses, pharmacists?

The requested information has been added to the abstract.

Line:35/36 affects the patient’s belief and perception of security about their medical treatment= referring back to the title do you mean beliefs about medical treatment of only medications specifically?

We specifically mean medication and the abstract has been updated accordingly. We have however chosen to use medication instead of medicines in this context. We believe the two words can be used interchangeably without confusing the reader. 

43 helpline, patient belief about medicine. Add (BMQ)

The abstract has been updated as suggested. 

43 helpline, patient belief about medicine-> patient’s beliefs

The abstract has been updated as suggested.

Line 44/45 Medication-related problems (MRPs) were identified and clinical impact of the MIH service was assessed. Of the person performing the service or during the telephone interviews?

The assessment was done after each case handling and it was done by a person that not involved in the specific case handling. We have tried to make this more clear in the abstract as proposed. 

47/48: thirty-seven questions were enquired by 26 participants to the MIH during the study-> please also add the percentage of how many had used the service and numbers from pre and post-trigger as well.

they all?n=26?

The abstract has been updated as suggested.

Line 48: Participants in the intervention group reported that the MIH increased their sense of security regarding medicine= all 119?

All 119 reported that the MIH increased their sense of security. However, the abstract has been updated based on input from another reviewer and this sentence has been deleted. 

From line 249: Both the control- and the intervention group responded that they felt safe to a great extent regarding their medication after discharge, and there was no significant difference between groups (p=0.59).

52 Conclusion: The MIH was highly appreciated by the participants, which indicate that the MIH is a valuable service for discharged patients in improving the sense of security in relation to medication and alleviating MRPs.

Please make this statement a little more modest based on the results from line 249.

We have updated the abstract and added the point from line 249 as suggested. Moreover, the conclusion Has been modified to a more modest statement.

Background

58: Transition from hospitals to private homes is known to challenge (the) patient safety. Remove the, and please add a reference.

The manuscript has been updated as suggested and a reference has been added.

Medication-related problems (MRPs), why MRPs and not DRP? How do you define the concept? You use DRPs in the conclusion.

We thank you for the comment and DRP has been changed to MRP in the conclusion. We have used the definition of MRP as stated in reference 3 and have added the definition to the manuscript.

Line 72: Medicine Information Helplines (MIH) can play an important role in supporting patients in a safe medication after hospitalization. Add a reference.

References have been added as requested.

74: A pharmacy professional. Wording, pharmacist?

Do other countries use MIH or only UK?

A sentence has been added providing information about MIH in other countries.

80: Patient may enquire the general practitioners or community pharmacies with medicine-related questions [9]. not the hospital?

We agree with the point and have corrected accordingly. 

85: This study is to our knowledge the first to explore if addition of MIH affects patient’s beliefs and perception of security about medicines as compared to standard care with limited medicine information support after discharge. Is this totally new or only in the Danish context? Please clarify.

To our knowledge this is totally new as not other studies have asked patients receiving standard care, thus, no control group have to our knowledge been reported.

I lack a summary of other possible services also aiming at supporting patients after discharge in regard to medication use. What makes MIH an alternative above other possible options?

We believe this has been accounted for in the manuscript: Patients may contact the general practitioners, or community pharmacies or the respective hospital ward with medicine-related questions [9]. However, this is not always suitable, as the new medicine may be used in hospitals only, the general practitioners may be unavailable, or the community pharmacy does not have access to the discharge summary from the hospital or the hospital staff might be busy taking care of admitted patients.

Method

Please start with the study design first.

Setting->? Description of the MIH?

Thank you for the comment. We believe that the setting section is more than a description of the MIH and that setting is appropriate.

Please describe the service in more in-depth. How many pharmacists offer the services, how were they trained in e.g. telephone consultations skills? Opening hours. Did some of them always carry a phone with them? Did you pilot the service first?

The manuscript have been updated with the suggested information.

Also briefly describe the hospital, its size?

Further details of the hospital have been added to the manuscript.

101: were recruited in the hospital wards-> in the two selected?

The manuscript has been updated accordingly.

Were it some inclusion criteria regarding the length of the stay at the ward? Did they have to use medications? Did they have to have had a medication change during the stay as this probably will impact their perceived information needs?

The length of stay or whether the patients used medicine or not were not exclusion criteria.

Study design

112 A randomized controlled study design was used to explore the study objective. Reference to a method book?

Reference to a method book has been added.

The included participants were randomized 1:4 (50:200). Why this ratio? Add a reference. I see you discussed this in the analysis section, maybe refer to that section?

The manuscript has been updated as suggested.

123 pharmacy technician from the MIC. Potential questions were documented. Only questions no actions/responses?

The manuscript has been updated as suggested.

135. and from a study by Badiani et 135 al. (2017) [7] were included in the interview guide. Is that something different from Hornes? Please describe. Add also what Danish version you used.

The interview guide in the current study included questions from both papers. The questions from Hornes are different from the ones form Badiani et la. We are not sure what is ment by adding the Danish version. If you find it relevant we will be happy to add the Danish version of the interview guide as supplementary information

136 further questions were included according to 136 the study objectives and contextual setting. Such as? Or refer to the appendix earlier.

All eligible participants were interviewed by telephone within 145 2-4 weeks after discharge. How did you decide upon the timeframe? Add an argument.

An argument has been added to the manuscript as suggested.

140: However, if a patient is not in any medical treatment prior to, during admission or after discharge, the patient is not able to provide an answer to each question from the interview guide.

In the current presentation of the data it has been accounted for if a patient has not been able to answer a question. Thus, for each question the number of answers are given. 

With medical treatment? Do you then mean medication use? If yes, please perform a sub-analysis with these persons excluded as this probably impacts your user rate of the service. 

A total of 3 patients from the intervention group were identified as not receiving no medication either at hospitalization nor at discharge. Comparison of data when removing the three patients did not result in any changes in the data shown in the manuscript.

I.e. do you have some data on which patients had some changes in their medication use, it would be interesting to see if these people more frequently used your service. If possible add this information, if not possible include it in the method discussion.

We thank the author for this comment. As shown in S3 Appendix there are two patients out the 8 patients who contacted the MIH who did not report a change in their medication. A total of 68 patients in the intervention group reported that they had a change in their medication. Of these 27 had a question regarding their medication (39,7%). 

A total of 18 patients in the control group reported that they had a change in their medication. Of these 4 patients had a question regarding their medication (22%). 

The data has been presented as results and discussed in the discussion. 

156 enquiry from the intervention group was determined individually by two clinical pharmacists. Why didn’t you include a medical doctor as well in this assessment?

We agree that the optimal setting would have been to include an assessment by both a pharmacist and a medical doctor. Due to work load pressure it was unfortunately not feasible to get the assessment done by any of the medical doctors involved in the current study.

162 No similar studies are available from which a sample size can be calculated. What about the studies from UK?

At the time of study design the authors were not aware of other studies using similar questionnaire and including a control group. Thus, we are not aware of other studies that can indicate an effect size We have to our best tried to set a sample size that can detect an 10% improvement on a 5-point scale as described in the manuscript. 

Results

How long were the consultations on average?

Information has been added to the manuscript. 

208 Enquiries to the MIH->MIH service

The manuscript has been updated as suggested.

Table 3

Explain the abbreviation MIH under the table.

Enquiries before

interview

Enquiries during

interview

Enquiries after

Interview

The table has been updated as suggested.

I don’t understand the categorization, do you mean during the consultation? What do you mean by after? Please add this information/categorization to your method section as well.

Enquiries during interview. Can you count this? I think you need to add these pre-post data in e.g. your abstract.

Further explanation has been added to the manuscript.

In the control group 55% of the participants (18/33) reportedly experienced a change in their medication during hospitalization, see S3 appendix. A similar level of change in medication was reported by participants in the intervention group (58%, 68/118), see S3 appendix. See previous comment. I’d like you to perform a subanalysis to see how this impacts the use of the service.

We have added further information on how big a percentage of the patients experiencing a change in medication also have a question to their medication and how many of the patients that used the MIH experienced to have a change in their medicines.

Line 233: Different formatting of the text

The different formatting is used because it is the legends to figure 2.

233 Patients scored satisfaction, with what?

A clarification has been added to the manuscript.

228 hospitalization and at discharge (figure 2).-> Figure 2.

The manuscript has been updated as suggested.

236 Approximately half of the participants (48 %, 73/141) preferred receiving their medicine information both written and orally. At discharge?

A clarification has been added to the manuscript. 

244:Among those, 67 % in both the control group (4/6) and intervention group (22/33) found an answer to their question by searching the internet or contacting their general practitioner, pharmacy, hospital, relatives or visiting nurses. Please refer to Appendix 3 for the distribution? N=X pharmacy.

A reference to Appendix 3 has been added.

246: in total, 8 patients from the intervention group contacted the 247 MIH and 100% (8/8). Didn’t you state: line 48: Thirty-seven questions were enquired by 26 participants to the MIH during the study,

The 8 patients contacted the MIH before the telephone interview. A clarification has been added to the manuscript.

263: Further, patients who contacted the MIH reported that the feeling of security increased after contacting the MIH, as they found the information reassuring and brought clarity to potential worries, such as side effects, dosage regimens etc. In addition, the patients informed that the helpline provided practical advice on how to administer the medication.

The patients contacting the MIH what characteristics did they have? What kind of beliefs about medicines did they have, did they differ from the others?

We thank you for the comment. We believe that all the information can be found in Appendix 3. We have included more details on the group of patients. 

267 Only patients in contact with the MIH before the telephone interview were asked about their satisfaction with the service (n = 8).?? I don’t understand, please elaborate on this. Why didn’t you ask the otherones?

We only executed the telephoneinterview once. This meant that all enqueries raised during or after the interview was answered after the actual telephone interview took place. In other words, patients that asked questions during or after the telephoneinterview were unable to assess satisfaction with the MIH.

The pharmacy interventions? What do you mean here?

We are not sure what this comment refers to.

314 Despite this a considerable number of patients still had questions regarding their medication after hospital discharge. Do you refer to the Appendix 3 question. I don’t agree as if I understand right only about 20% contacted the MIH 26/119=21%

Do you know why not more contacted your service of the 28% having questions?

We have modified the statement. Our point is that event the respondents express a high level of satisfaction with the medicine information received still18- 28% had questions to their medicines. As described in the manuscript the respondents found answers in different ways like the internet, GP, pharmacy, relatives or visiting nurse. We did not ask them why they did not choose the MIH as first place to search for information.

Discussion.

I lack a discussion about how you would like to improve the MIH offered and a comparison to other similar services. In general, what outcomes do you think are important to assess? If someone wants to perform a similar service that would be of interest.

We believe that we already have provided a discussion on this as the literature shows that if you actively follow-up. We have on this suggestion added a clarification to the discussion. 

316/317: Patients who were offered access to the helpline reported an increase in their feeling of security in relation to their medication. Yes, but the groups did not differ regards beliefs about medicines etc. Please add this information.

The manuscript has been revised as suggested.

323 The present study shows that patients have a need for information about their medication after discharge, more specifically, 26 % of the participants in the current study. 26/119?=21%? Or which numbers do you refer to? Is it question 6?

It is correct that we here refer to the total number of answers given to question 6. The numbers are clearly stated in the result section. We have however revised this part of the discussion according to input from reviewer 1. We hope the reference to the data is now more clear.

333 MIH is limited in this study, it should be seen in the context that the service is newly established to patients. As the MIH becomes a well-known service, we anticipate that the number of enquiries increase over time. So, you don’t think you need some more personal follow up to increase the numbers? Do you have some data from the patients on this what might make them contact MIH?

We have no data on what might make them contact the MIH. We have provided further input to the discussion regarding follow-up contact which has been proven effective in several studies. 

340: Studies indicate that providing the appropriate level of medicine information after discharge may reduce medicine related rehospitalization or contact to primary medical doctor or physician [1, 4, 23, 24]. In this context a MIH may not only improve the patient safety of the medication after discharge but it may also decrease overall costs of treatment. However, further studies are needed to elucidate the cost-benefit of a MIH service to patients after discharge. Do you have some data on how many also had tried to contact their doctor and you parallel?

None of the 8 patients that contacted the MIH prior to the telephone interview contacted their GP. For the remaining patients we don’t know whether they contacted parallel. Unfortunately, we did not collect these data.

350 with the current study, Badiani et al. found that 93% of the respondents followed the advice provided in their study [7]. How do you know this that they followed your advice?

The data supporting this statement are from question 21-24 – see S3 Appendix.

361 Although patients in both groups reported that they to a high degree felt safe with their medication after discharge, the current study revealed that patients with access to the MIH expressed an increase in the feeling of security in relation to their medication. Do your data support this claim? Please reframe.

We believe that our data support this statement as we specifically aske patients in the intervention group on this issue – see question 14 in S3 Appendix. We have provided a reference to S3 Appendix to the statement and rephrased the sentence.

379: The study did not include the full Beliefs About Medicine Questionnaire [14], thus not allowing a fully valid conclusion on patients concerns and necessity of taking medicine. Why did you not include that?

We fully agree that it would have been highly relevant to include the full BMQ. However, at the time of study design we found that the final questionnaire became too long and took away focus on the evaluation of the MIH. Furthermore at study by the University of Copenhagen (not published yet) had experience in assessing patient concerns only using part of the questions included in the BMQ. For these reasons we decided to only use selected questions from the BMQ.

383 A methodological consideration in the current study is the risk of respondent’s social desirability. Who did perform the interviews?

A description of who performed the interviews is described in the Methods section. The telephoneinterviews were preformed by author 1, 2 and 3.

Other important outcomes to assess in a future study in addition to adherence?

Outcomes as patient motivation and barriers to contact an MIH has been added to the discussion. 

Conclusion

405 MIH offers an important support for discharged patients to alleviate DRPs. Here you use DRPs not MRPs please be consistent and that the service positively affects a safe medication after discharge. Might impact.

The conclusion has been revised as suggested.

Please make sure you answer your aims in the conclusion. Also add information about how many used the service pre/post prompt.

We believe that the conclusion answers the aims: satisfaction with MIH, clinical significance of intervention and effect on patients feeling of safety in relation to medication.

405 MIH offers an important support for discharged patients to alleviate DRPs and increase the belief and perception of security with their medicine after discharge. Can you draw that conclusion? The groups appear to have similar beliefs about medicines and level of security.

We agree with the reviewer and have revised the conclusion.

---

## [Decision Letter · Decision Letter 1]

1 Aug 2023

PONE-D-22-33407R1Medicine Information Helpline after Hospitalization - a randomzed trial: impact on patient satisfaction, patient concerns about medicines and clinical outcome on patient safetyPLOS ONE

Dear Dr. Clemmensen,

Thank you for submitting your manuscript to PLOS ONE. After careful consideration, we feel that it has merit but does not fully meet PLOS ONE’s publication criteria as it currently stands. Therefore, we invite you to submit a revised version of the manuscript that addresses the points raised during the review process.

The revised manuscript addresses major concerns raised by reviewers on the prior version. Address points raised by reviewer#2 for clarity sake. 

We look forward to receiving your revised manuscript.

Kind regards,

Vineet Gupta, MD, FACP, SFHM

Academic Editor

PLOS ONE

Journal Requirements:

Reviewers' comments:

Reviewer's Responses to Questions

**Comments to the Author**

1. If the authors have adequately addressed your comments raised in a previous round of review and you feel that this manuscript is now acceptable for publication, you may indicate that here to bypass the “Comments to the Author” section, enter your conflict of interest statement in the “Confidential to Editor” section, and submit your "Accept" recommendation.

Reviewer #2: (No Response)

Reviewer #4: All comments have been addressed

2. Is the manuscript technically sound, and do the data support the conclusions?

Reviewer #2: Yes

Reviewer #4: Partly

3. Has the statistical analysis been performed appropriately and rigorously? 

Reviewer #2: Yes

Reviewer #4: Yes

4. Have the authors made all data underlying the findings in their manuscript fully available?

Reviewer #2: Yes

Reviewer #4: No

5. Is the manuscript presented in an intelligible fashion and written in standard English?

Reviewer #2: Yes

Reviewer #4: Yes

6. Review Comments to the Author

Reviewer #2: Minor revisions:

1- Sample size estimate: State the statistical testing method which achieves 90% power.

2- Define SD at its first appearance.

3- The Wilcoxon rank sum test is the complete name of the test.

4- Table 2: Remove repeat p-values for individual characteristics. For instance, place the p-value of 0.540 on the line for "Sex, No. (%)" rather than on the lines for Male and Female. Also relabel this characteristic "Gender".

5- Figure 2 caption: Clarify that "No significance" refers to "No significant differences".

6- In the "Data analysis" section, state the level of the p-value that has been used to define statistically significant differences. For example many publications include the following sentence: P-values less than 0.05 were considered statistically significant.

7- Patient satisfaction with MIH: The first sentence of this paragraph suffers from subject-verb disagreement. The corrected sentence reads, "One of the primary outcomes was patient satisfaction with MIH."

8- For primary outcomes reported as percentages, provide 95% confidence intervals for these percentages. State the statistical method used to estimate these confidence intervals in the "Data analysis" section of the manuscript.

9- Discussion: Subject-verb disagreement correction: "As expected, the rate of patients who had a question regarding their medicines was higher ..."

10- To assist in the review process, add line numbering to the document.

Reviewer #4: (No Response)

7. PLOS authors have the option to publish the peer review history of their article (what does this mean?). If published, this will include your full peer review and any attached files.

Reviewer #2: No

Reviewer #4: No

---

## [Author Response · Author response to Decision Letter 1]

21 Sep 2023

Thank you for giving us the opportunity to revise the manuscript. We thank you for your review which have given us a chance to further improve the manuscript.

Changes to reference list

There have been no changes to the reference list. Thus, the submitted reference list is identical to the one submitted previously. 

Reviewer 2: 

1- Sample size estimate: State the statistical testing method which achieves 90% power.

No statistical testing method was used to achieve a 90% power. The power was set by the authors to 90% which we consider to be a conservative choise as most studies will accept a power of 80%. A small clarification has been added to the data analysis to clarify that the level of power was set by the authors (line 193-194). 

2- Define SD at its first appearance.

Thank you for the comment. We have added an explanation to “SD” when it is first introduced in the manuscript (line 202). 

3- The Wilcoxon rank sum test is the complete name of the test.

The manuscript has been updated accordingly.

4- Table 2: Remove repeat p-values for individual characteristics. For instance, place the p-value of 0.540 on the line for "Sex, No. (%)" rather than on the lines for Male and Female. Also relabel this characteristic "Gender".

We thank the reviewer for the comment and has updated Table 2 as suggested. 

5- Figure 2 caption: Clarify that "No significance" refers to "No significant differences".

The Figure 2 caption has been revised as suggested. 

6- In the "Data analysis" section, state the level of the p-value that has been used to define statistically significant differences. For example many publications include the following sentence: P-values less than 0.05 were considered statistically significant.

The manuscript has been updated with the suggested wording in the Data analysis section (line 193-194).

7- Patient satisfaction with MIH: The first sentence of this paragraph suffers from subject-verb disagreement. The corrected sentence reads, "One of the primary outcomes was patient satisfaction with MIH."

Thank you for this review. The manuscript has been updated accordingly.

8- For primary outcomes reported as percentages, provide 95% confidence intervals for these percentages. State the statistical method used to estimate these confidence intervals in the "Data analysis" section of the manuscript.

The primary outcomes are reported as follows: 

- Patient satisfaction with the MIH is reported as a mean score and the CI can be found in S7 Supporting Information, Question 25.

- Patients feeling of safety regarding their medication is assessed by Question 9-14. Question 9-13 is reported as mean score and the CI can be found in S7 Supporting Information. Question 14 is reported as percentage and a confidence interval has been added to the manuscript regarding question 14 (line 289). Furthermore, a sentence has been added to the method section describing the method used to calculate the CI (line 203).

- Clinical impact of MIH is reported with exact numbers (descriptive statistics) and thus a CI is not calculated. 

9- Discussion: Subject-verb disagreement correction: "As expected, the rate of patients who had a question regarding their medicines was higher ..."

Thank you for the input. The manuscript has been corrected. 

10- To assist in the review process, add line numbering to the document.

Line numbering has been added to the document.

---

## [Editor Report · Decision Letter 2]

16 Oct 2023

Medicine Information Helpline after hospitalization – a randomized trial: impact on patient satisfaction, patient concerns about medicines and clinical outcome on patient safety

PONE-D-22-33407R2

Dear Dr. Clemmensen,

We’re pleased to inform you that your manuscript has been judged scientifically suitable for publication and will be formally accepted for publication once it meets all outstanding technical requirements.

Kind regards,

Vineet Gupta, MD, FACP, SFHM, CHCQM

Academic Editor

PLOS ONE
---

## [Editor Report · Acceptance letter]

19 Oct 2023

PONE-D-22-33407R2 

Medicine Information Helpline after hospitalization – a randomized trial: impact on patient satisfaction, patient concerns about medicines and clinical outcome on patient safety 

Dear Dr. Clemmensen:

I'm pleased to inform you that your manuscript has been deemed suitable for publication in PLOS ONE. Congratulations! Your manuscript is now with our production department. 

Kind regards, 

on behalf of

Dr Vineet Gupta 

Academic Editor

PLOS ONE